# Classification of Color Fundus Photographs Using Fusion Extracted Features and Customized CNN Models

**DOI:** 10.3390/healthcare11152228

**Published:** 2023-08-07

**Authors:** Jing-Zhe Wang, Nan-Han Lu, Wei-Chang Du, Kuo-Ying Liu, Shih-Yen Hsu, Chi-Yuan Wang, Yun-Ju Chen, Li-Ching Chang, Wen-Hung Twan, Tai-Been Chen, Yung-Hui Huang

**Affiliations:** 1Department of Information Engineering, I-Shou University, No. 8, Yida Road, Jiao-Su Village, Yan-Chao District, Kaohsiung City 84001, Taiwan; 2Department of Medical Imaging and Radiological Science, I-Shou University, No. 8, Yida Road, Jiao-Su Village, Yan-Chao District, Kaohsiung City 82445, Taiwan; 3Department of Radiology, E-DA Cancer Hospital, I-Shou University, No. 21, Yida Road, Jiao-Su Village, Yan-Chao District, Kaohsiung City 82445, Taiwan; 4School of Medicine for International Students, I-Shu University, No. 8, Yida Road, Jiao-Su Village, Yan-Chao District, Kaohsiung City 84001, Taiwan; 5Department of Life Sciences, National Taitung University, No. 369, Sec. 2, University Road, Taitung City 95048, Taiwan; 6Institute of Statistics, National Yang Ming Chiao Tung University, No. 1001, University Road, Hsinchu 30010, Taiwan

**Keywords:** color fundus photographs, CNN, deep learning

## Abstract

This study focuses on overcoming challenges in classifying eye diseases using color fundus photographs by leveraging deep learning techniques, aiming to enhance early detection and diagnosis accuracy. We utilized a dataset of 6392 color fundus photographs across eight disease categories, which was later augmented to 17,766 images. Five well-known convolutional neural networks (CNNs)—efficientnetb0, mobilenetv2, shufflenet, resnet50, and resnet101—and a custom-built CNN were integrated and trained on this dataset. Image sizes were standardized, and model performance was evaluated via accuracy, Kappa coefficient, and precision metrics. Shufflenet and efficientnetb0demonstrated strong performances, while our custom 17-layer CNN outperformed all with an accuracy of 0.930 and a Kappa coefficient of 0.920. Furthermore, we found that the fusion of image features with classical machine learning classifiers increased the performance, with Logistic Regression showcasing the best results. Our study highlights the potential of AI and deep learning models in accurately classifying eye diseases and demonstrates the efficacy of custom-built models and the fusion of deep learning and classical methods. Future work should focus on validating these methods across larger datasets and assessing their real-world applicability.

## 1. Introduction

Eye diseases represent a significant health concern worldwide, with a growing prevalence that poses substantial challenges to healthcare systems and impacts patients’ quality of life. Common ocular conditions such as age-related macular degeneration, cataracts, glaucoma, and diabetes-related eye diseases, among others, necessitate early detection and accurate diagnosis to improve treatment outcomes and reduce disease burden. However, the classification of these conditions using color fundus photographs—a common diagnostic tool in ophthalmology—remains a challenging task due to factors such as class similarities, data imbalance, image quality and resolution variability, and the potential presence of multiple concurrent diseases. Emerging technologies such as Artificial Intelligence (AI) and Machine Learning (ML) have shown great potential in addressing these challenges by processing large datasets, identifying subtle patterns, and making accurate predictions. In particular, convolutional neural networks (CNNs), a type of deep learning model, have been demonstrated to perform exceptionally well in image classification tasks [1]. Their ability to automatically and adaptively learn spatial hierarchies of features makes them a suitable candidate for the analysis of medical images, including color fundus photographs.

Zhang et al. present an innovative model that jointly optimizes a cycle generative adversarial network (CycleGAN) and a convolutional neural network (CNN) to detect retinal diseases and localize lesions with limited training data [2]. The authors have leveraged the capabilities of CycleGAN for generating more realistic images, and a novel res-guided sampling block combining residual features and pixel-adaptive convolutions adds to the uniqueness of this model. The model’s successful performance with the LAG, Ichallenge-PM, and Ichallenge-AMD datasets indicates its potential for effective disease classification and localization. The study suggests that the joint optimization network can yield highly competitive results, further emphasizing the promising future of AI in medical imaging—making it an insightful review that explores the promising capabilities of AI in enhancing the detection and management of glaucoma. It successfully highlights the potential of AI in enhancing clinical decision-making and improving the efficiency and quality of glaucoma care. The discussion on future perspectives, especially on the role of AI in expanding access to glaucoma care worldwide, is particularly exciting and thought-provoking [3]. The focus on different diagnostic modalities for glaucoma, including fundus photography, OCT imaging, standard automated perimetry, and gonioscopy, adds depth to the paper. This provides readers with a comprehensive overview of how AI can be utilized across various diagnostic tools, enhancing the paper’s value for clinicians, researchers, and even industry experts [4,5,6,7]. Moreover, some studies could have significant implications for eye health diagnostics, specifically in estimating axial length and SFCT, traditionally performed by optical coherence tomography (OCT) and optical low-coherence reflectometry. The automated nature of this method can improve efficiency and potentially enable more widespread access to these types of assessments, which could be particularly beneficial in resource-limited settings [8,9]. From a clinical perspective, insights into the morphology of the choroidal vessel layer can be valuable in the diagnosis and differentiation of these conditions, which share certain symptoms and are often confused with one another. However, they each have different treatment approaches and prognoses, emphasizing the importance of accurate diagnosis [10]. A potential limitation of this study could be its reliance on existing diagnostic imaging technologies, which may not provide a complete picture of the intricate changes occurring at the choroidal level. However, the use of advanced imaging modalities such as optical coherence tomography angiography (OCTA) could overcome this limitation, offering a more detailed view of the choroidal vasculature [11,12].

One study emphasizes that the application of CNN model-based screening could significantly reduce the burden on ophthalmologists at reading centers, which is crucial in high-demand healthcare systems. This, along with the reduction of potential waiting times for patients and the ability to scale screening procedures, is an essential step forward in enhancing the efficiency of healthcare delivery [13]. However, while these results are encouraging, further research is needed to optimize these models and validate their use in diverse, real-world clinical settings. It is also essential to assess the cost-effectiveness and the practicality of integrating such AI systems into existing healthcare infrastructures [13,14,15]. Shi et al. focus on the development of a deep learning model known as Med-XAI-Net, designed to detect the presence or absence of geographic atrophy (GA) from OCT volume scans. The need for such a system is clear, given the complex and time-consuming nature of manual GA identification. The model’s purpose was not only to automate GA detection but also to make AI decisions more interpretable for practitioners by highlighting the most relevant B-scan regions showing GA [16]. These results suggest that the use of such AI models could significantly improve the efficiency and accuracy of GA detection in clinical practice. By guiding ophthalmologists to the most relevant scans and regions, such models can save time and potentially improve patient outcomes by aiding earlier diagnosis and treatment.

This study aims to further explore this potential by developing and testing deep learning models for the classification of eye diseases using a dataset of color fundus photographs. The paper presents the implementation and comparison of several well-known CNN architectures, along with a custom-designed CNN model. We also evaluate the impact of various data augmentation techniques to counter class imbalance and enrich the dataset, aiming to enhance model performance. By establishing the effectiveness of CNN models in the classification of eye diseases, this study intends to contribute to the existing literature on AI’s role in ophthalmology and the broader field of medicine. It also aims to lay the groundwork for further studies on the integration of these advanced technologies into routine clinical practice, potentially transforming the field of ophthalmological care.

This paper is organized as follows: Section 1 provides an introduction to the study, detailing its significance and outlining our key objectives. Section 2 comprehensively reviews the related works, with a particular focus on the classification of fundus photographs employing machine learning and deep learning techniques. In Section 3, we elaborate on the materials and methodologies employed for this research. We outline our research flow, the datasets used, the image processing techniques implemented, and the application of transferred learning in five distinct CNN models. This section also provides an introduction to the fusion of extracted features from transferred learning CNNs with SVM for enhanced classification, the creation of a customized CNN model, and the performance metrics used for model evaluation. Section 4 is dedicated to showcasing the results obtained from the application of the various methods mentioned in Section 3. Following this, in Section 5, we delve into a discussion of these results, analyzing the implications, strengths, and weaknesses of the adopted approaches and comparing these results with those of previous studies. Finally, Section 6 concludes the paper, summarizing the critical findings and contributions of our research and offering a perspective on potential future directions.

The major contributions of this study are as follows.
We present a comprehensive comparison of the performance of five renowned convolutional neural networks (CNNs)—efficientnetb0, mobilenetv2, shufflenet, resnet50, and resnet101—in the classification of eye diseases from color fundus photographs. To the best of our knowledge, such a comprehensive evaluation of these specific models in the context of eye disease classification has not been conducted previously.We introduce a novel custom-built CNN model optimized for the task of eye disease classification from fundus photographs. The architecture of this model, comprising 17 layers, was specifically designed to tackle the challenges posed by this classification task.We demonstrate the effectiveness of various data augmentation techniques in enhancing model performance and addressing class imbalance—a prevalent issue in medical imaging datasets.We investigate the utility of a fusion approach that combines the image features extracted by deep learning models with classical machine learning classifiers. Our study illustrates the significant potential of this technique in enhancing the accuracy of eye disease classification.This research makes a significant contribution by validating the efficiency and effectiveness of deep learning models in the classification of eye diseases, thereby contributing to the literature on AI’s role in ophthalmology and broader healthcare. The study serves as a groundwork for future research focusing on the integration of these advanced technologies into routine clinical practice.Finally, we conduct a rigorous assessment of model performance using accuracy, Kappa coefficient, and precision metrics. This rigorous evaluation lends robustness to our findings and provides a basis for comparing our models with those proposed in future studies.

## 2. The Related Works of Classification Fundus Photographs

There are a number of articles indicating a robust and active research interest in the application of deep learning methods for the classification of eye conditions using fundus photographs. The publications cover a variety of topics, including the use of convolutional neural networks (CNNs) to detect and classify various ocular diseases like diabetic retinopathy, glaucoma, age-related macular degeneration, and retinopathy of prematurity. This reflects the broad potential of deep learning in enhancing the automated analysis of fundus photographs. Many of the studies reported high diagnostic accuracy, suggesting that deep learning models could support ophthalmologists in making more precise and faster diagnoses, improving patient management. Some papers also address the interpretability of these models, a crucial aspect of integrating AI tools in clinical practice. However, the application of deep learning in this field is not without challenges. Some articles highlight the need for large, high-quality, and diverse datasets for training models. Concerns about model generalizability across different populations and imaging devices also exist. Some papers emphasize the need for further clinical validation and regulatory considerations before these AI tools can be fully integrated into routine clinical practice. Hence, while the use of deep learning for fundus photograph classification is a promising and rapidly evolving field, continued research and development are necessary to address the existing challenges and maximize the potential benefits of these technologies in ophthalmology.

This paper demonstrates an exciting development in the use of deep learning for the detection of diabetic retinopathy (DR) from retinal fundus photographs. The results are highly promising, with the algorithm achieving a sensitivity of over 90% and specificity of over 93% at two different operating points in detecting referable diabetic retinopathy (RDR) [17]. One paper has designed a deep learning approach based on the deep residual neural network (resnet101) for the automatic detection of glaucomatous optic neuropathy (GON) using color fundus images [18]. A notable aspect of this study is the occlusion testing, which demonstrated that the model identified the neuroretina rim region and retinal nerve fiber layer (RNFL) defect areas as the most crucial for GON discrimination, mimicking the approach of a human clinician. An artificial intelligence (AI) system was designed to predict high myopia grades derived from Optical Coherence Tomography (OCT) based on fundus photographs [19]. By training a novel deep learning model using a large set of qualified fundus photographs, the researchers were able to detect and predict myopic maculopathy according to the atrophy (A), traction (T), and neovascularization (N) classification and grading system. The deep learning model demonstrated impressive accuracy, with an area under the receiver operating characteristic curve (AUC) of 0.969 for category A, 0.895 for category T, and 0.936 for category N. The average accuracy across categories was between 85.34% and 94.21%. Moreover, the performance of the AI system was superior to that of attending ophthalmologists and comparable to that of retinal specialists. The developed AI system may serve as a valuable tool for predicting vision-threatening conditions in high myopia patients using simple fundus photographs. This could potentially reduce the cost of patient follow-up and expand accessibility to diagnostic support in underdeveloped areas that only have access to fundus photography. However, further validation and real-world application testing of the system are necessary to confirm its potential benefits [19,20,21].

Moreover, a deep learning ensemble model was proposed to automatically grade the severity of glaucoma stages using fundus photographs. The final dataset consisted of 3460 fundus photographs from 2204 patients, categorized into three classes: unaffected controls, early-stage glaucoma, and late-stage glaucoma. They trained 56 convolutional neural networks (CNNs) with various characteristics and developed an ensemble system to combine several modeling results for optimal performance [22]. It demonstrated an accuracy of 88.1% and an average area under the receiver operating characteristic (AUC) of 0.975, outperforming the best single CNN model, which achieved an accuracy of 85.2% and an AUC of 0.950. However, as with any AI-based diagnostic tool, it is crucial to validate the model’s performance in real-world clinical settings before deployment.

Therefore, in this study, implementing a classification system for eight classes using fusion extracted features and customized convolutional neural network (CNN) models for color fundus photographs can pose several challenges as followings.

Data Collection: The first challenge would be the collection of a sufficient amount of high-quality, labeled fundus photographs for each of the eight classes. Each class needs a substantial amount of data to ensure that the CNN can learn the distinguishing features accurately.

Data Quality: Fundus photographs can vary greatly in quality due to differences in imaging equipment, lighting conditions, the presence of artifacts, and other factors. Poor image quality could make it difficult for the CNN to learn and predict accurately.

Data Imbalance: There may be a class imbalance problem where some classes have many examples, and others have few. This can negatively impact the performance of the model as it can cause the model to be biased towards the classes with more examples.

Feature Extraction: Customizing a CNN to extract relevant features from fundus images for each class is a complex task. It requires deep knowledge and understanding of both the underlying disease pathology and how these features manifest in fundus photographs.

Multi-class Classification: The difficulty of the task increases with the number of classes. Distinguishing between eight classes is more difficult than a binary classification problem, as it introduces more opportunities for misclassification.

Interclass Variation and Intraclass Similarity: If there are high degrees of variation within a class (intraclass variation) and high degrees of similarity between different classes (interclass similarity), it can make the classification problem significantly more challenging.

Despite these challenges, with appropriate data, resources, and techniques, it is possible to develop a reliable and accurate multi-class classification model using fundus photographs and deep learning.

## 3. Materials and Methods

### 3.1. The Flow of Research

The research flowchart is defined as input images, image augmentation, organized training strategies, training 3 kinds of CNN Models, evaluated performance, and obtained final results (Figure 1).

The summary of these steps is proposed, and illustrations as below.

Input Images: The initial step is to collect and prepare a dataset of color fundus photographs, each labeled with one of the eight classes.

Image Augmentation: This step involves applying a normal distribution to simulate different levels of intensity noise in a given image (Y) to augment the dataset. This can help improve the robustness and generalization of the model.

Organizing Training Strategies include Setting Hyper-Parameters: Adjusting various hyperparameters, such as batch size, number of epochs, image resizing parameters, and learning rates; Splitting Data: Separating the data into a training set (70%) and a testing set (30%); Selecting CNN Models with Transfer Learning: Five pre-trained models—efficientnetb0, mobilenetv2, shufflenet, resnet50, and resnet101—will be used for transfer learning; User-Designed CNN Model: Designing a custom CNN model with 17 layers.

Training Three Types of CNN Models include Model 1: Transfer learning with the five pre-selected CNN models; Model 2: A fused model that merges features extracted from the five pre-selected CNN models, followed by classification using a Support Vector Machine (SVM); Model 3: The custom-designed 17-layer CNN model.

Evaluation of Performance: Evaluate the models using various performance metrics, including accuracy, recall, precision, and the Kappa statistic.

Results: Analysis and interpretation of the performance of the three types of models, including identifying the most effective model(s) and discussing any significant findings. This step may also involve fine-tuning the models or revisiting previous steps based on the evaluation results.

The proposed methodology in this study combines transfer learning, feature fusion, and a custom deep learning architecture, which offers a comprehensive approach to classify fundus photographs into eight classes. The findings have the potential to contribute to the advancement of machine learning techniques in ophthalmological imaging and diagnosis.

### 3.2. The Datasets

The study used a publicly available dataset from Kaggle (URL: https://www.kaggle.com/datasets/andrewmvd/ocular-disease-recognition-odir5, accessed on 5 June 2023), consisting of 6392 color fundus photographs categorized into eight distinct disease classes (Figure 2): age-related macular degeneration (A), cataract (C), diabetes (D), glaucoma (G), hypertension (H), pathological myopia (M), normal (N), and other diseases/abnormalities (O). Due to the inherent class imbalance in the dataset, we employed data augmentation techniques to enrich the dataset, resulting in a total of 17,766 images with a more balanced distribution across all categories (Table 1). The images were RGB in JPG format.

Our methodology incorporated the use of five pre-established convolutional neural network (CNN) architectures—efficientnetb0, mobilenetv2, shufflenet, resnet50, and resnet101, along with a bespoke CNN model. These models were trained and tested using the enriched dataset of fundus photographs. Hyperparameter tuning was performed to optimize each CNN model’s learning process and performance, with parameters such as batch size, epochs, optimizer, and learning rates carefully adjusted. Given computational constraints and to enhance model efficiency, all images were resized to two standard dimensions: 64 × 64 pixels.

The performance of each model was thoroughly evaluated using three metrics: accuracy, Kappa coefficient, and precision. This rigorous assessment approach allowed for a holistic understanding of each model’s performance and the overall effectiveness of our methodology.

The schematic of data augmentation utilized in our study is shown in Figure 3. This technique is a good way to augment data without the need for collecting new data, which can be expensive and time-consuming, especially for medical imaging where patient consent, privacy, and logistics are major concerns. Through this data augmentation process, we simulate different noise conditions, thereby enhancing the robustness of the model to potential noise in real-world data. It is important to note that the effectiveness of this data augmentation strategy can depend on several factors, including the nature of the noise added, the distribution of the noise, the original data, and the specific task. Therefore, it is always important to empirically validate the effectiveness of the chosen data augmentation strategy with training or test data.

### 3.3. Image Processing

This study implemented two essential steps in image preprocessing to facilitate accurate analysis. Firstly, we loaded the fundus image data into memory, transforming it into a suitable format for subsequent evaluation. This step encompassed resizing the images to a matrix size of 64 × 64 and normalizing the pixel intensity values to fall within the range of 0 to 1.

### 3.4. Transferred Learning for Five CNNs

In this study, we employed transfer learning with five pre-established convolutional neural network (CNN) architectures: efficientnetb0, mobilenetv2, shufflenet, resnet50, and resnet101. The use of transfer learning enabled us to leverage these pre-trained models’ learned feature extraction capabilities, providing an excellent starting point for our task of image classification. Our training strategy commenced with the careful setting of key hyperparameters. These included defining the appropriate batch size, the number of training epochs, the dimensions for image resizing, and learning rates. Proper adjustment of these hyperparameters was crucial to ensure the efficiency of the learning process and the ultimate performance of our models. To ensure an unbiased evaluation of the models, we adopted a standard data-splitting strategy: 70% of the images were allocated for training the models, and the remaining 30% were reserved for testing. This partitioning ensures that the models were evaluated on previously unseen data, providing a reliable measure of their potential performance in real-world applications.

The selected CNN architectures were then employed with the transfer learning approach. The models, initially trained on large image datasets like ImageNet, had proven capabilities for extracting complex features from images. By using these models as the starting point and retraining them on our specific task, we hoped to obtain robust and reliable image classifiers. Therefore, this study represents a comprehensive application of transfer learning using five different CNN architectures for image classification. This methodology allowed us to exploit the strengths of each model, setting the stage for robust and reliable classification performance.

### 3.5. Fusion Extracted Features from Transferred Learning CNNs with Classifiers

The second approach to our classification task involved the fusion of features extracted from the five CNN architectures trained using the transfer learning approach. By combining the features learned by each model, we aimed to encapsulate a wider range of discriminatory information, which could lead to improved classification performance. After training the CNN models and obtaining the output from each network’s penultimate layer, we proceeded to concatenate these high-dimensional features to form a combined feature vector. This amalgamation of features from different models leveraged the unique strengths of each individual network, creating a more comprehensive representation of the input images. To make use of these fused features, we employed a Support Vector Machine (SVM), Logistic Regression (LR), and Naïve Bayes (NB) for the final classification task. SVM, LR, and NB have been shown to perform well in high-dimensional spaces and are known for their robustness to overfitting, making them an excellent choice for this task.

The model, henceforth referred to as Model 2, followed a two-step process. First, it utilized the combined power of five different CNN architectures to extract complex, high-level features from the input images. Subsequently, SVM, LR, and NB classifiers were used to perform the final classification based on these fused features. In conclusion, this hybrid approach enabled us to harness the feature extraction capabilities of multiple CNN models, coupled with the robustness and high-performance characteristics of these classifiers. We anticipate that this approach would lead to enhanced classification performance by utilizing a wider range of image features.

### 3.6. A Customized CNN Model

In this study, we sought to extend the capabilities of pre-existing CNN architectures by developing a customized CNN model specifically tailored to our task of classifying eight categories of fundus photographs. This model, referred to as Model 3, is a 17-layered deep learning architecture designed to capture complex hierarchical features from the images. Herein, we present the architecture of Model 3.

This user-designed CNN model incorporates proven strategies from deep learning research, such as convolutional layers for feature extraction, ReLU activations for non-linearity, max pooling for dimensionality reduction, and batch normalization for accelerating learning. By customizing these elements to our specific task, we aim to achieve superior classification performance. Meanwhile, this user-designed CNN model combines powerful deep learning techniques to create an architecture that is suitable for the specific task at hand, potentially leading to superior performance (Table 2). This carefully designed architecture aims to extract intricate patterns and structures from the image data. Each layer contributes to a gradual abstraction of features, from the raw input image to high-level features that allow for accurate classification. The unique combination of operations in our model is anticipated to provide superior results in the classification of eye diseases. Here is a detailed breakdown of the model architectures below.

### 3.7. Performance Index for Classification

The evaluation of the classification performance of convolutional neural networks (CNNs) and other machine learning models often involves several metrics, providing a multi-faceted understanding of the model’s effectiveness. In this context, we used the recall rate, precision, accuracy, and Cohen’s Kappa values for model assessment.

Recall Rate: Also known as sensitivity or true positive rate, the recall rate quantifies the proportion of actual positive cases accurately identified by the model. It is particularly critical in scenarios where minimizing false negatives is paramount, such as medical diagnoses where a missed condition could lead to serious repercussions. The recall rate is computed as the ratio of True Positives (TP) to the sum of True Positives and False Negatives (FN) (Equation (1)).
Recall Rate = TP/(TP + FN)(1)

Precision: The precision, or positive predictive value, signifies the percentage of positive identifications by the model that are indeed positive. This metric becomes crucial when the ramifications of false positives are high, such as in spam detection, where misclassifying legitimate messages can be problematic. Precision is computed as the ratio of True Positives to the sum of True Positives and False Positives (FP) (Equation (2)).
Precision = TP/(TP + FP)(2)

Accuracy: Accuracy provides a generalized measure of the model’s performance by calculating the ratio of correct predictions (both positive and negative) over all predictions. While useful, accuracy can be misleading in imbalanced datasets, where predicting the majority class can result in superficially high accuracy (Equation (3)).
Accuracy = (TP + TN)/(TP + TN + FP + FN)(3)

Cohen’s Kappa: The Kappa statistic measures the statistical agreement between the model’s predictions and the actual labels, taking into account any agreement that occurs by chance. It is a more robust metric than accuracy in handling class imbalance. A higher Kappa value signifies better classification performance, with 1 representing perfect agreement and 0 denoting agreement purely by chance (Equation (4)).
Kappa = (Observed Accuracy − Expected Accuracy)/(1 − Expected Accuracy)(4)

The F1-Score is the harmonic mean of precision and recall. Unlike the arithmetic mean, the harmonic mean tends toward the smaller of the two elements. Therefore, if either precision or recall is low, the F1-Score will also be low. This makes it a useful metric when you need to take both precision and recall into account. It is worth noting that the F1-Score is most useful in situations where you have an uneven class distribution. If false positives and false negatives are equally important to your context, you may want to consider using balanced accuracy or another similar metric (Equation (5)).
F1-Score = 2 × (Precision × Recall)/(Precision + Recall)(5)

By employing these metrics, we can gain a comprehensive understanding of the model’s performance. This facilitates model selection, hyperparameter tuning, and informed decision-making when deploying the model in real-world applications.

## 4. Results

In the presented model 1, the performances of five different convolutional neural networks (CNNs) after transfer learning, trained with three different optimizers, are shown in Table 3. The accuracy (Acc) and Kappa coefficient (Kappa) are listed for each combination of CNN and optimizer. These metrics provide insights into the efficiency of each combination in the classification task. Table 2 suggests that the combination of efficientnetb0 and Adam optimizer provided the best performance, achieving an accuracy of 0.713 and a Kappa value of 0.672. In contrast, the combination of resnet50 and the SGDM optimizer had the lowest performance, with an accuracy of 0.438 and a Kappa value of 0.358.

The performance of the efficientnetb0model was assessed across eight disease classes using a range of metrics, including precision, recall, F1-Score, overall accuracy, and Kappa coefficient (Figure 4). For age-related macular degeneration (Class A), the model demonstrated impressive efficacy, achieving a recall of 90.6%, precision of 88.7%, and F1-Score of 89.6%. In distinguishing cataract cases (Class C), the model also performed exceptionally well, with a high recall of 97.3%, precision of 86.3%, and an F1-Score of 91.5%.

However, the model struggled to correctly classify diabetes (Class D), evident from a considerably low recall of 3.7%, precision of 37.8%, and an F1-Score of just 6.8%. For glaucoma (Class G), the model showed robust performance, garnering a recall of 93.4%, precision of 78.4%, and an F1-Score of 85.3%. Hypertension (Class H) was identified with near-perfect accuracy by the model, as indicated by the high recall rate of 98.2%, precision of 97.0%, and an F1-Score of 97.6%. Pathological myopia (Class M) was also predicted with high accuracy by the model, demonstrated by a recall of 97.9%, precision of 95.3%, and an F1-Score of 96.6%. However, the model had difficulties in accurately predicting normal cases (Class N), reflected by a low recall of 15.2%, precision of 53.1%, and an F1-Score of 23.7%. For the identification of other diseases and abnormalities (Class O), the model achieved a decent recall of 77.8%, though precision was low at 34.2%, leading to a moderate F1-Score of 47.5%. Hence, the model demonstrated relatively good performance with an accuracy of 71.3% and a Kappa statistic of 0.672. The Kappa statistic is a reliable indicator of the model’s agreement with the actual labels, taking into account random chance. The score of 0.672 suggests substantial agreement, showcasing the reliability of the model. However, lower performance in identifying diabetes and normal cases indicates areas that require improvement in future model iterations.

Table 4 presents the performance of classification using extracted features from five CNNs after transfer learning, coupled with different classifiers (Model 2). The performance is assessed with varying batch sizes and evaluated based on the accuracy (Acc) and Kappa coefficient (Kappa). Here, three classifiers are considered: Logistic Regression (LR), Naive Bayes (NB), and Support Vector Machine (SVM).

Table 4 indicates that Logistic Regression exhibited the best overall performance with a maximum accuracy and Kappa coefficient of 0.821 and 0.796, respectively, achieved at a batch size of 9. While performance slightly varied with different batch sizes, the results remained consistently high. In comparison, Naive Bayes and Support Vector Machines yielded slightly lower values across the board. This table suggests that, within the scope of these tests, the fusion of features extracted from pre-trained CNNs coupled with a Logistic Regression classifier offers superior performance.

Following the application of the Logistic Regression (LR) on the fusion of features, the results presented in the confusion matrix reveal the performance of this model across the eight classes of fundus photographs (Figure 5). For class A, the model demonstrated high accuracy, with a recall rate of 98.0% and a precision of 98.2%. This suggests only 2.0% of the instances were misclassified (False Negative, FN), and 1.8% of the predicted class A images were incorrect (False Positive, FP). The F1-Score, which is the harmonic mean of precision and recall, was noted to be 98.1%. Class C also had a remarkable performance, with the highest recall and precision rates of 99.7% and 98.3%, respectively. The F1-Score for this class was equally impressive at 99.0%. Contrastingly, classes D and N had comparatively lower performances. For class D, the recall and precision rates were 53.0% and 50.8%, respectively, resulting in an F1-Score of 51.9%. Similarly, class N reported a recall and precision of 50.0% and 59.7%, respectively, translating to an F1-Score of 54.4%. Among the remaining classes, classes G, H, and M each displayed high recall rates of 99.6%, 99.9%, and 99.6%, respectively, as well as strong precision rates of 98.0%, 99.5%, and 99.7%, respectively. The F1-Scores were subsequently reported to be 98.8%, 99.7%, and 99.7% for classes G, H, and M, respectively. Finally, class O reported a modest performance with a recall rate of 61.1%, a precision of 55.7%, and an F1-Score of 58.3%. The model, which utilized the fusion of features with logistic regression, had an accuracy of 82.5% and a Cohen’s Kappa of 0.800, signifying substantial agreement. These results suggest that the fusion of features from multiple convolutional neural networks (CNNs) combined with a Logistic Regression classifier provides a robust solution for the multi-class classification of fundus images. It is worth noting that classes D and N might require further optimization for improved model performance.

Table 5 reports the performance of a user-designed 17-layer CNN model tested over multiple runs, utilizing a batch size of 50 and 25 epochs (Model 3). The results are measured in terms of accuracy (Acc) and the Kappa coefficient (Kappa). It can be observed that the user-designed CNN demonstrates a high level of performance, achieving a maximum accuracy and Kappa coefficient of 0.930 and 0.920, respectively. Despite slight variations in results across different runs, the performance remains consistently high, thereby validating the efficiency and effectiveness of the user-designed model. This further suggests that the customized 17-layer architecture is well-suited for the classification task at hand.

The performance metrics of our user-designed convolutional neural network (CNN) model for the multi-class classification of fundus images are depicted in the table. These metrics encompass recall, precision, accuracy, Kappa, and F1-Score for each class (A, C, D, G, H, M, N, O), as well as the overall performance of the model (Figure 6).

Class A demonstrates a recall of 95.5%, precision of 99.4%, and F1-Score of 97.4%. Class C follows closely with a recall of 98.9%, precision of 97.3%, and F1-Score of 98.1%. Class D, however, experienced a noticeable drop in recall (83.3%) but maintained a high precision of 82.9%, resulting in an F1-Score of 83.1%. For classes G, H, and M, the CNN model performed remarkably well, with recall values of 98.5%, 99.2%, and 99.5%, respectively. This translated to F1-Scores of 97.4%, 98.4%, and 99.2% for the same classes, demonstrating a strong concurrence between precision and recall rates. For class N, the recall was 83.2%, with a precision of 82.5%, resulting in an F1-score of 82.9%. Lastly, for class O, our model achieved a recall of 87.7% and a precision of 91.1%, leading to an F1-Score of 89.4%.

Our CNN model achieved an overall accuracy of 93.2% and a Kappa statistic of 0.922, underscoring the model’s efficacy in classifying fundus images across multiple classes. The relatively high precision, recall, and F1-Scores across the classes indicate the robustness of the proposed CNN model.

## 5. Discussion

The comparative performance of different models and methods employed in the study is summarized in Table 6. This presents a benchmark comparison between the various strategies employed for ocular disease classification.

Model 1 incorporates transfer learning techniques using five different convolutional neural networks (CNNs). Of these, efficientnetb0 achieved the highest accuracy (0.713) and Kappa coefficient (0.672). Shufflenet, resnet50, resnet101, and mobilenetv2 followed, with respective accuracies and Kappa values as displayed.

In Model 2, a fusion of features extracted from the five pre-trained CNNs was used in conjunction with Logistic Regression (LR) as a classifier. This approach resulted in the best performance of the three models, with an accuracy of 0.821 and a Kappa value of 0.796.

Finally, a customized CNN was designed for Model 3, consisting of 17 layers. This user-designed model had a solid performance with the highest accuracy and Kappa value across all the methods tested (accuracy: 0.930, Kappa: 0.920). These results provide insights into the effectiveness of each approach for ocular disease classification using fundus photographs. The comparison further emphasizes the potential benefits of developing custom CNNs over using transferred learning techniques or fusion feature strategies.

The presented method was benchmarked against related works in the field, and the results are presented in Table 7 [23,24,25,26,27,28,29,30,31]. The most recent study by [23] utilized a Hybrid CNN-RNN with Artificial Humming Bird Optimization, achieving an accuracy of 97.4% on binary classification. A similar level of accuracy was seen in the 2021 study by Huang H, Wang X, Ma H [25] that used a Feature Pyramid Network and Vovnet with an accuracy of 98.6%. The highest accuracy (99.1%) was obtained by Yang HK, et al. [29] in 2020, employing deep learning image recognition in three classes.

In comparison, the presented method uses a user-designed CNN model and addresses the classification of eight classes, making it a more complex task. Our method achieved an accuracy of 93.0%, which is the highest among the studies dealing with more than two classes, signifying the effectiveness of our approach in multi-class classification scenarios. Despite the more challenging task, our methodology maintains competitive performance, indicating a promising direction for future research in ocular disease classification.

The user-designed CNN was specifically tailored for the task at hand, taking into consideration the unique aspects of the data. This custom fit likely led to improved results, as the model was optimized to identify patterns specific to the given task. It is important to note that while the user-designed CNN performed best in this particular task, it may not necessarily be the best for all tasks. The performance of a model can depend on a range of factors, including the nature of the data, the specific task, and the model’s architectural and training parameters. Therefore, it is always crucial to evaluate a model’s performance using a suitable validation scheme and fine-tune or modify the model as necessary based on the evaluation results.

Moreover, while the study has several strengths, including a novel approach, rigorous evaluation, and potential practical applications, it also has limitations that should be addressed in future work. This might include using larger and more varied datasets, further validating the results, improving the interpretability of the model, and optimizing the computational efficiency.

## 6. Conclusions

This study illustrates the effectiveness of deep learning approaches in the classification of ocular diseases using color fundus photographs. Both transferred learning and fusion image features with classifiers have demonstrated promising accuracies and Kappa coefficients, underlining their potential as valuable tools in disease identification. Moreover, the user-designed 17-layer convolutional neural network (CNN) showed superior performance, with accuracies reaching up to 93% and Kappa of 0.92. The performance of transferred learning methods highlighted the utility of leveraging pre-existing neural network architectures, such as shufflenet, resnet50, resnet101, efficientnetb0, and mobilenetv2. These models, originally trained on large-scale datasets, provided a robust foundation for the task of ocular disease identification. In the case of fusion image features with classifiers, the results affirmed the value of combining deep learning extracted features with classical machine learning classifiers. Logistic regression (LR) showcased the best performance among the evaluated classifiers. The user-designed CNN has further validated the capacity of deep learning in processing medical imaging data. Despite its complexity, the model was able to achieve impressive results, confirming the worthiness of customizing deep learning architectures to specific tasks. This study, however, was not without limitations. These included potential overfitting risks, the requirement of large and diverse training datasets, and certain inflexibility in the models’ structures. These findings provide insight into areas that future studies could focus on improving.

In the future, we aim to validate our model using an extensive dataset and diverse types of ocular images. We also plan to incorporate a multi-modal learning approach by integrating fundus images with clinical parameters to enhance diagnostic performance.

## Figures and Tables

**Figure 1 healthcare-11-02228-f001:**
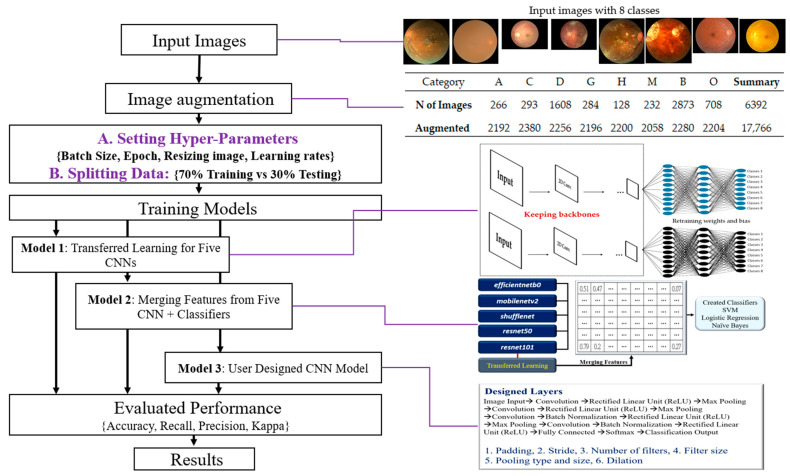
The flowchart demonstrates the presented methods followed in this study.

**Figure 2 healthcare-11-02228-f002:**
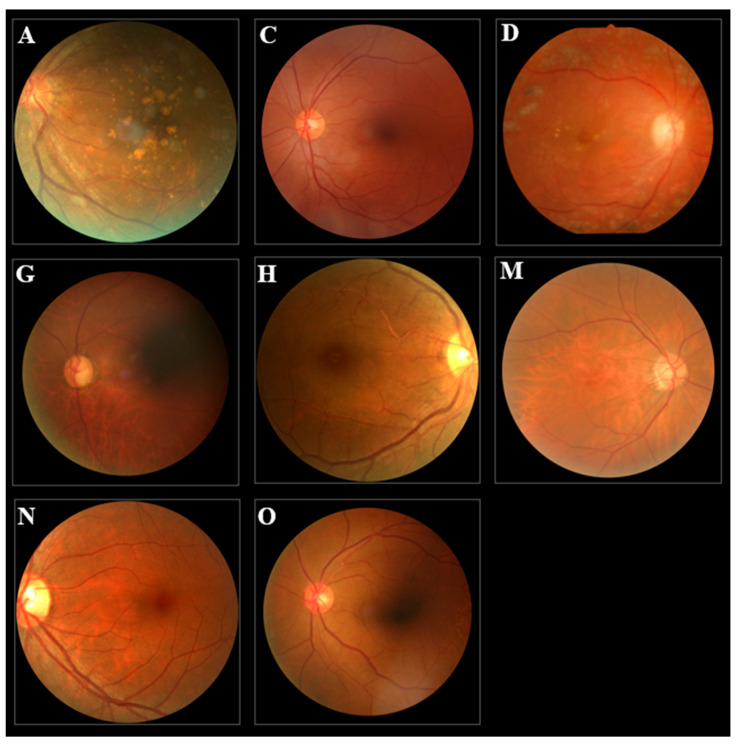
The example images of eight groups, including age-related macular degeneration (A), cataract (C), diabetes (D), glaucoma (G), hypertension (H), pathological myopia (M), normal (N), and other diseases/abnormalities (O).

**Figure 3 healthcare-11-02228-f003:**
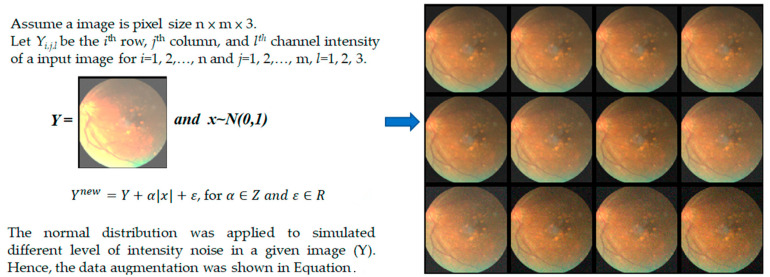
The schema of data augmentation was shown in this study.

**Figure 4 healthcare-11-02228-f004:**
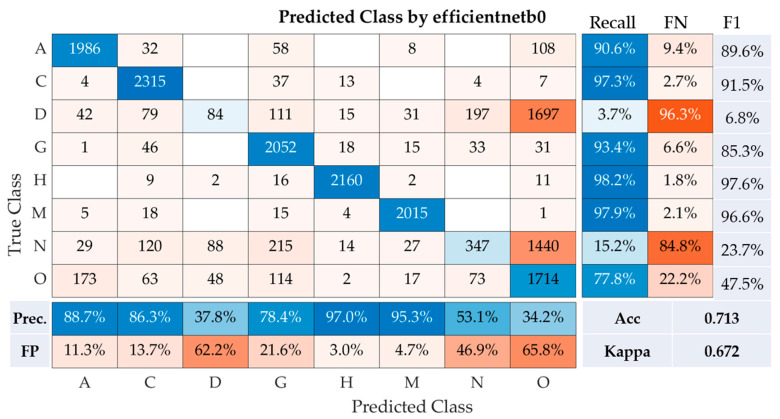
The confusion matrix generated by the efficientb0. Note: Prec is precision, Acc is accuracy, FN is false negative rate, and FP is false positive rate.

**Figure 5 healthcare-11-02228-f005:**
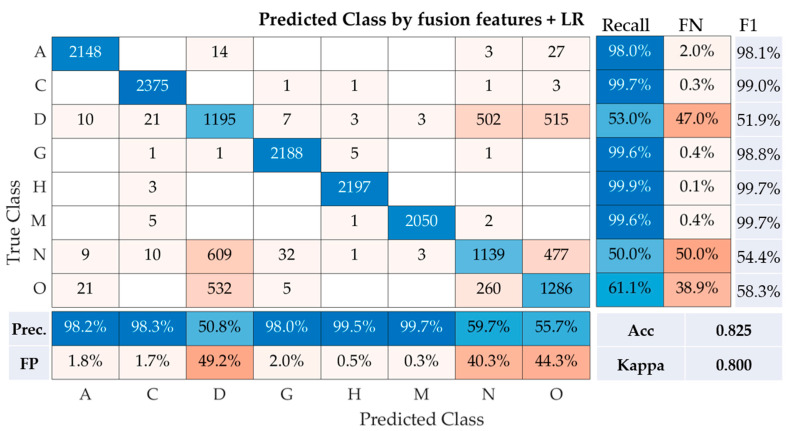
Performance metrics of the Logistic Regression model utilizing fusion features for multi-class classification of fundus images. Note: Prec is precision, Acc is accuracy, FN is false negative rate, and FP is false positive rate.

**Figure 6 healthcare-11-02228-f006:**
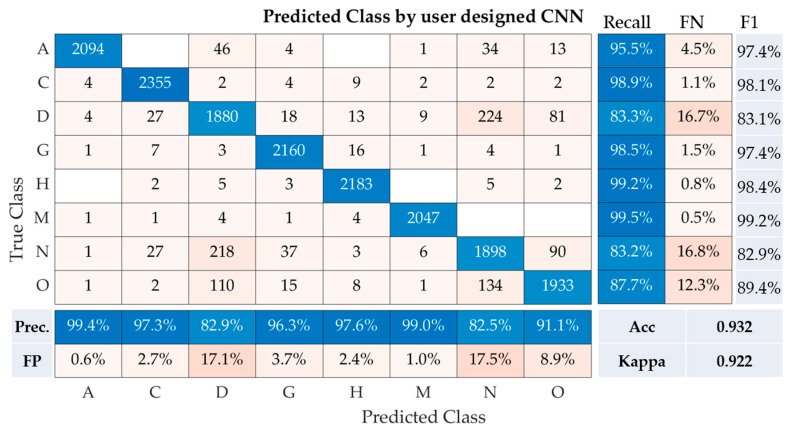
Performance metrics of the user-designed CNN model for multi-class classification of fundus images. Note: Prec is precision, Acc is accuracy, FN is false negative rate, and FP is false positive rate.

**Table 1 healthcare-11-02228-t001:** The numbers of color fundus photograph, age of patient, counts of gender, and numbers of images after augmented.

Category	Female	Male	Female Age Mean ± SD	Male Age Mean ± SD	N of Images	Augmented Images
Age-related Macular Degeneration (A)	125	141	60.6 ± 12.6	61.7 ± 10.5	266	2192
Cataract (C)	168	125	67.3 ± 10.8	66.1 ± 13.2	293	2380
Diabetes (D)	692	916	56.7 ± 10.3	54.9 ± 10.1	1608	2256
Glaucoma (G)	118	166	65.4 ± 9.4	61.4 ± 11.7	284	2196
Hypertension (H)	47	81	60.9 ± 6.1	54.8 ± 7.4	128	2200
Pathological Myopia (M)	157	75	52.4 ± 24.0	60.1 ± 11.4	232	2058
Normal (N)	1305	1568	58.6 ± 11.3	56.0 ± 11.1	2873	2280
Other diseases/abnormalities (O)	356	352	61.4 ± 9.9	57.8 ± 11.2	708	2204
Summary	2968	3424	59.0 ± 12.2	56.8 ± 11.2	6392	17,766

**Table 2 healthcare-11-02228-t002:** The architectures of user-designed CNN model.

Layer	Descriptions
1	Image Input Layer: 32 × 32 × 3 images were fed into the network with ‘zerocenter’ normalization to standardize pixel values across images.
2	Convolutional Layer 1: This layer applies 256 filters of size 3 × 3, using a stride of [1 1] and padding ‘same’ to maintain the spatial dimensions of the input.
3	Activation Layer 1: This layer applies the Rectified Linear Unit (ReLU) activation function, a popular choice in CNNs for introducing non-linearity.
4	Max Pooling Layer 1: A 2 × 2 max pooling operation is performed with a stride of [2 2] and padding of [0 0 0 0] to reduce the spatial dimensions and increase the receptive field size.
5	Convolutional Layer 2: Similar to the first convolutional layer, this layer applies 512 filters of size 5 × 5, using stride [1 1] and ‘same’ padding.
6	Activation Layer 2: This layer also applies the ReLU activation function.
7	Max Pooling Layer 2: A 2 × 2 max pooling operation is performed with a stride of [2 2] and padding of [0 0 0 0], further reducing spatial dimensions.
8	Convolutional Layer 3: This layer applies 1024 filters of size 3 × 3, using stride [1 1] and ‘same’ padding.
9	Batch Normalization Layer 1: This layer normalizes the activations of the previous layer to accelerate learning and improve model stability.
10	Activation Layer 3: The ReLU activation function is applied.
11	Max Pooling Layer 3: A 2 × 2 max pooling operation is conducted with a stride of [2 2] and padding of [0 0 0 0].
12	Convolutional Layer 4: This layer applies 2048 filters of size 9 × 9, using stride [1 1] and ‘same’ padding.
13	Batch Normalization Layer 2: This layer further normalizes the activations of the previous layer to speed up learning and provide stability to the network.
14	Activation Layer 4: The ReLU activation function is applied.
15	Fully Connected (Dense) Layer: This layer contains 8 neurons, corresponding to the eight classes of fundus diseases.
16	Activation Layer 5: This layer applies the Softmax activation function, which is used for multi-class classification. It converts the output to probability scores, providing a final classification.
17	Classification Output Layer: This layer uses a cross-entropy loss function to guide the learning process of the network, driving the model’s predictions closer to the true labels.

**Table 3 healthcare-11-02228-t003:** The performance of classification using five CNNs after transferred learning (Model 1).

CNN	Optimizer	Acc	Kappa
shufflenet	sgdm	0.655	0.606
shufflenet	adam	0.691	0.647
shufflenet	rmsprop	0.681	0.635
resnet50	sgdm	0.438	0.358
resnet50	adam	0.682	0.637
resnet50	rmsprop	0.627	0.574
resnet101	sgdm	0.468	0.391
resnet101	adam	0.679	0.633
resnet101	rmsprop	0.598	0.541
efficientnetb0	sgdm	0.560	0.496
efficientnetb0	adam	0.713	0.672
efficientnetb0	rmsprop	0.627	0.574
mobilenetv2	sgdm	0.570	0.508
mobilenetv2	adam	0.661	0.613
mobilenetv2	rmsprop	0.670	0.623

**Table 4 healthcare-11-02228-t004:** The performance of classification using extracted features from five CNNs after transferred learning with classifiers (Model 2).

Classifier	Batch	Acc	Kappa
LR	3	0.819	0.794
LR	6	0.819	0.793
LR	9	0.825	0.800
LR	12	0.820	0.794
LR	15	0.821	0.795
NB	3	0.784	0.753
NB	6	0.783	0.751
NB	9	0.782	0.751
NB	12	0.783	0.752
NB	15	0.783	0.752
SVM	3	0.783	0.752
SVM	6	0.789	0.759
SVM	9	0.795	0.765
SVM	12	0.800	0.771
SVM	15	0.786	0.755

**Table 5 healthcare-11-02228-t005:** Performance metrics of the user-designed 17-layer CNN with batch size 50 and 25 epochs (Model 3).

Batch	Epoch	Acc	Kappa
50	25	0.932	0.922
50	25	0.924	0.913
50	25	0.924	0.913
50	25	0.918	0.907
50	25	0.908	0.895
50	25	0.907	0.893
50	25	0.906	0.893
50	25	0.900	0.885
50	25	0.859	0.838
50	25	0.853	0.832

**Table 6 healthcare-11-02228-t006:** Comparative performance of different models and methods used for ocular disease classification.

Model	Method	Classifier	Acc	Kappa
1	Transferred Learning	shufflenet	0.691	0.647
		resnet50	0.682	0.637
		resnet101	0.679	0.633
		efficientnetb0	0.713	0.672
		mobilenetv2	0.670	0.623
2	Fusion Features	LR	0.821	0.796
3	User-Designed Layers	CNN	0.930	0.920

**Table 7 healthcare-11-02228-t007:** The results of the presented method were compared to related works.

Author(s)	Year	Method	Classes	Accuracy (%)
Dhiravidachelvi E, et al. [23]	2023	Hybrid CNN-RNN with Artificial Humming Bird Optimization	2	97.40
Gómez-Valverde JJ, et al. [24]	2019	Convolutional Neural Networks and Transfer Learning	2	87.48
Huang H, Wang X, Ma H [25]	2021	Feature Pyramid Network and Vovnet	2	98.60
El-Den NN, et al. [26]	2023	Scale-Adaptive Auto-Encoder-Based Model with Deep Learning	2	96.20
van Grinsven MJ, et al. [27]	2016	CNN with Selective Data Sampling	2	87.40
Beeche C, et al. [28]	2023	Neural Understanding Network (NUN)	3	91.10
Yang HK, et al. [29]	2020	Deep Learning Image Recognition	3	99.10
Wang K, et al. [30]	2023	Convolutional Neural Networks with Self-Attention	8	88.70
Mateen M, et al. [31]	2022	Hybrid Feature Embedding Approach	2	96.00
The Presented Method	2023	User-Designed CNN	8	93.00

## Data Availability

The ocular disease intelligent recognition (ODIR) data set was acquired from Kaggle URL: https://www.kaggle.com/datasets/andrewmvd/ocular-disease-recognition-odir5k (accessed on 5 June 2023).

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
