# Peer review of "Classification of Color Fundus Photographs Using Fusion Extracted Features and Customized CNN Models"

_healthcare, 2023, doi:10.3390/healthcare11152228_

Round 1

Reviewer 1 Report

This article utilizes deep learning technology to challenge the classification of eye diseases, aiming to improve the accuracy of early detection and diagnosis. Overall, the novelty of this article is limited, and the motivation and challenges have not been well defined. Here are some comments that I hope can improve the quality of the manuscript.

1.The abstract section provides a brief overview of the research, but does not fully summarize the main findings of the paper. I hope the author can modify the abstract to provide a clearer summary of the results.

2. It is recommended to list the contributions made in this article through the point 1.2.3.4... after the introduction.

3.Chapter 3.6 describes the network you proposed, but there is no detailed introduction in the text. It is only a few paragraphs of description that are not intuitive. Please provide a detailed supplement.

4. There are too few images in the article, and there is no network structure diagram or experimental result diagram. It is recommended that the author use the form of a diagram to express it more intuitively.

5. More ablation and comparative experiments should be added to highlight the functionality of each structure. The comparison points of ablation experiments in the article are relatively single, and there are few evaluation indicators for the comparative experiments.

6. We believe that network models should be used for other types of image data to enhance the generalization of the model.

7. Further clinical pathological analysis of experimental results is needed to assist in medical diagnosis.

8. Remember to add a discussion at the end of this article to point out the future improvement directions.

Please further correct the sentences in the article to make it more logical.

Reviewer 2 Report

Dear Authors,

Thank you for working on this important project. The study generally well described and results are well presented. I have a few comments that can significantly improve your manuscript:

1. Clearly describe your data: Provide images from you dataset and in some cases you have augmented these images and made them RGB, show those converted images using Figures. That will help provide a better understanding of how the data was converted. Provide code for those conversions (on Github) so that others can recreate those images. 

1 (a) Data augmentation should be done only for training data, did you augment data for all training and test sets as well?

1 (b) I probably missed this but did you use X-ray images as well? What are the different kinds of images you used. Provide clear descriptions. 

2. Provide a detailed architecture of your new model in a Figure. What were the kernel sizes? Provide every details like you see for RESNET 50 or any other pretrained architecture in the original papers. 

3. Why did your model perform better? Explain this well. This should be a part of your discussion.

4. What were some of the strengths and limitations of your study? Describe those.

5. Typo: Fix the first sentence of conclusion- In this study has illustrated

Additional considerations:

1. Perform calibrations for your models

2. Consider adding loss and accuracy plots for train and test data

3. Add details of fine tuning if you performed any.  

Reviewer 3 Report

In the manuscript entitled “Classification of Color Fundus Photographs Using Fusion Extracted Features and Customized CNN Models” the authors aimed to deep nueral network based model for classification of eye diseases employing 6392 color fundus photographs across 8 disease categories. The work seems to be significant in the biomedical context and this reviewer thinks that it can be published with minor revisions.  My concerns are as follows

1. Authors should clearly explain the original source of datasets (Primary source/research work done along with Kaggle), how different models are deployed? (links to the code such as GitHub, Codeocean etc) and visualization of CNN model, performance metrices (in the supplementary file or as link).

2. Elaborate the potential applications and future implications of this work.  

Round 2

Reviewer 1 Report

After modification, the overall quality of this paper has been improved.